# Methyl Group Metabolism in Differentiation, Aging, and Cancer

**DOI:** 10.3390/ijms23158378

**Published:** 2022-07-29

**Authors:** Lars Erichsen, Chantelle Thimm, Simeon Santourlidis

**Affiliations:** 1Institute for Stem Cell Research and Regenerative Medicine, Medical Faculty, Heinrich-Heine University Düsseldorf, 40225 Düsseldorf, Germany; chantelle.thimm@uni-duesseldorf.de; 2Epigenetics Core Laboratory, Institute of Transplantation Diagnostics and Cell Therapeutics, Medical Faculty, Heinrich-Heine University Düsseldorf, Moorenstr. 5, 40225 Düsseldorf, Germany; simeon.santourlidis@med.uni-duesseldorf.de

**Keywords:** methyl group metabolism, cellular differentiation, aging, cancer

## Abstract

Methyl group metabolism belongs to a relatively understudied field of research. Its importance lies in the fact that methyl group metabolic pathways are crucial for the successful conversion of dietary nutrients into the basic building blocks to carry out any cellular methylation reaction. Methyl groups play essential roles in numerous cellular functions such as DNA methylation, nucleotide- and protein biosynthesis. Especially, DNA methylation is responsible for organizing the genome into transcriptionally silent and active regions. Ultimately, it is this proper annotation that determines the quality of expression patterns required to ensure and shape the phenotypic integrity and function of a highly specialized cell type. Life is characterized by constantly changing environmental conditions, which are addressed by changes in DNA methylation. This relationship is increasingly coming into focus as it is of fundamental importance for differentiation, aging, and cancer. The stability and permanence of these metabolic processes, fueling the supplementation of methyl groups, seem to be important criteria to prevent deficiencies and erosion of the methylome. Alterations in the metabolic processes can lead to epigenetic and genetic perturbations, causative for diverse disorders, accelerated aging, and various age-related diseases. In recent decades, the intake of methyl group compounds has changed significantly due to, e.g., environmental pollution and food additives. Based on the current knowledge, this review provides a brief overview of the highly interconnected relationship between nutrition, metabolism, changes in epigenetic modifications, cancer, and aging. One goal is to provide an impetus to additionally investigate changes in DNA methylation as a possible consequence of an impaired methyl group metabolism.

## 1. Introduction

The mammalian life cycle begins with the fertilization of the oocyte, which starts proliferating rapidly and thereby giving rise to the whole organism with hundreds of different cell types. This process is referred to as embryogenesis and required several subsequent steps of differentiation. During embryogenesis, the only totipotent cell is the fertilized oocyte [1], with its unique ability to specialize into all extraembryonic as well as all embryonic tissues and organs [2]. During the following cell divisions, first cell fate decisions are made. At the stage of the blastocyst, two different lineages with divergent differentiation capacity are established. These are the trophoblast, giving rise to the extra embryonal structures, and the inner cell mass (ICM). By subsequent asymmetrical divisions, the ICM is further divided into primitive ectoderm (epiblast) and primitive endoderm (hypoblast) in the late blastocyst stage [1]. The epiblast gives rise to pluripotent stem cells forming the entire embryo which are referred to as embryonic stem cells. With each subsequent division, a gradual loss of differentiation capacity is acquired leading ultimately to a fully differentiated somatic cell. Despite their completely different differentiation capacity, all of the before-mentioned cells in humans have in common, that they all contain the same 3.2-billion base pairs of DNA [3] harboring 20,687 protein-coding genes [4]. So how is the different function and phenotype of all cells within an organism established since all cells share the same genetic information?

Particular mechanisms that are needed for the selective expression of genes are referred to as epigenetics and were first postulated by Conrad Hal Waddington [5]. As already mentioned, each cell of an organism shares the same genetic information but requires a specific regulation of gene expression to attain the respective functional and structural properties. In the 1980s, it was described that the general levels of DNA methylation in the cells of an early developing embryo are relatively low compared to fully differentiated somatic cells [6]. Furthermore, in 1992, Kafri et al. showed that at various genes of an oocyte or in the sperm a “resetting” of the epigenetic code takes place [7]. During the later stages of development, the resetted sequences again get methylated in a process that is referred to as “de novo” methylation, executed by DNA methyltransferases 3a and 3b (DMNT3a/b) [8]. These processes of resetting and “de novo” methylation both contribute to the determination of different cell fates during embryogenesis [9] and are able to repress genes through DNA methylation or activate tissue-specific genes by demethylation [8]. Once these changes are established, they become automatically maintained during each cell division, even in the absence of the originally initiating factors. Responsible for maintaining the DNA methylation pattern during each cell division is DNA methyltransferase 1 (DNMT1).

The aim of this review is to provide a compact overview of the diverse methyl group metabolic pathways operating in each cell to provide the required methyl groups for acquiring and maintaining the appropriate phenotype and function. In addition, we want to highlight that deterioration of these metabolic pathways may be key to an accelerated aging process and the development of aging-related diseases. Finally, we want to discuss some natural compounds and nutrients, which can interfere with the metabolism of methyl groups in humans and thereby have the potential to slowdown the aging process and provide anti-cancer activity. 

## 2. Methyl Group Metabolism

There are two major metabolic pathways regulating the supplementation of methyl groups within a mammalian cell: (1) one-carbon metabolism and (2) polyamine metabolism.

### 2.1. The One-Carbon Metabolism

One-carbon metabolism is composed of the folate cycle, methionine cycle (METZ), and the transsulfuration pathway. One-carbon metabolism is characterized by a series of cyclic reactions, in which each time a one-carbon group is transferred. Besides the function of providing methyl groups for all methylation reactions (including DNA, RNA, and proteins), one-carbon metabolism is also essential for the production of phospholipids and nucleotides during proliferation [10]. In addition, one-carbon metabolism can regulate the cellular redox status through the oxidation of NADPH and the generation of glutathione [11].

### 2.2. Folate Cycle

The folate cycle occurs in both the cytosol and the mitochondria and starts with the uptake of folate with nutrition [12]. Folate is the water-soluble form of vitamin B9 and is brought into the cytosol in its reduced form tetrahydrofolate (THF) by folate receptors. THF is converted to 5,10-methylene-THF by serine hydroxymethyltransferases 1/2 (SHMT1/2), and this happens either in the cytosol or the mitochondrion. For this reaction, mainly serine and to a lesser extent glycine serve as carbon group donors [13,14]. In the next step, 5,10-methylene-THF is converted into 5-methyl-THF by methylenetetrahydrofolate reductase (MTHFR). This reaction only happens in the cytosol and 5-methyl-THF is used in the methionine cycle for the remethylation of homocysteine to methionine. In contrast, the 5,10-methylene-THF in the mitochondrion is converted into 5,10-methenyl-THF and 10-formyl-THF. 10-formyl-THF can only be transferred out of the mitochondrion in this form and can then be used for the majority of carbon-dependent reactions in the cytoplasm and nucleus [15,16]. These include, besides the generation of metabolites used in the folate and methionine cycle, also contributions to purine synthesis [12]. An overview of the folate cycle is given in Figure 1.

The folate cycle is carried out in the cytosol as well as in the mitochondria. It begins with the uptake of folate through the diet. Folate is the water-soluble form of vitamin B9 which is taken up via folate receptors in the reduced form of THF. It can be converted to 5,10-methylene-THF by serine hydroxy methyltransferase in the cytosol and in mitochondria. Primarily, serine and to a lesser extent glycine serve as carbon group donors in this process. Further, the 5,10-methylene-THF can then be converted by methylenetetrahydrofolate reductase into 5-methyl-THF, which is used for the remethylation of homocysteine to methionine. In the cytosol and nucleus, it can then supply carbon groups to the majority of “one-carbon” dependent reactions.

### 2.3. Methionine Cycle

The methionine cycle starts with the remethylation of homocysteine (HCY) to methionine (MET) by methionine synthase (MTR). For this reaction, the 5-methyl-THF, produced in the folate cycle by MTHFR, and vitamin B12 as a cofactor are required [11]. In some tissues, such as, e.g., the kidneys or the liver, the remethylation reaction can also be carried out by other enzymes such as betaine-homocysteine-S-methyltransferase (BHMT) [17]. In this reaction, 5-methyl-THF is replaced by betaine as a methyl group donor. Subsequently, under ATP consumption, the adenylation of the formed MET leads to the formation of S-adenosyl-L-methionine (SAM), which serves as a central molecule fueling all methylation reactions, including DNA and histone methylation, by donation of a methyl group [10]. This reaction is carried out by the family of methionine-adenosyltransferases (MATs). MET and folate both can enter the human body with our nutrition, with half of the MET being directly converted into SAM [18]. This emphasizes the importance of proper nutrition to establish and maintain a functional epigenome. The process of methylation is performed by histone methyltransferases (HMTs) or DNA methyltransferases (DNMTs). For both reactions, the methyl group of the SAM is either transferred to a lysine or arginine residue of a histone protein or to the 5′-carbon atom of the pyrimidine base cytosine, producing S-adenosyl-homocysteine (SAH) [11]. The re-entry of SAH into the METZ is achieved by the hydrolysis of the molecule into HCY and adenosine. This hydrolysis is carried out by the enzyme adenosylehomocysteinase (AHCY). With the remethylation of HCY to MET, which is carried out by methionine synthase (MTR), the METZ is completed [10]. The main methyl group donor for the remethylation of HCY to MET is serine. An overview of the METZ is given in Figure 2.

The methionine cycle begins with the remethylation of homocysteine to methionine. This reaction requires the 5-methyl-THF produced in the folate cycle by MTHFR and vitamin B12 as a cofactor. The recycling of the B12 cofactor by MTHFR represents a key reaction of remethylation. The subsequent adenylation of methionine leads to the formation of SAM, which serves as a methyl group donor for DNA and histone methylation. This reaction is catalyzed by MAT. MET, just like folate, can be supplied to the body with food, whereby half of the MET is converted directly into SAM.

### 2.4. Transsulfuration Pathway

Another pathway besides the METZ, in which HCY is metabolized, is the transsulfuration pathway. In fact, approximately 60% of the cellular HCY is mined in this cycle [19]. The reactions in this metabolic pathway are irreversible and require vitamin B6 as a cofactor for the enzymatic activity of the initiating enzyme cystathionine ß-synthase (CBS). This enzyme catalyzes the conversion of HCY to cystathione. In the further steps of the cycle, cystathione is converted into cysteine and ultimately to glutathione. In contrast to folate and MET, HCY cannot be taken up by nutrition. It must be synthesized through metabolic processes in the human body itself. An overview of the transsulfuration pathway is given in Figure 3.

Homocysteine can be degraded not only through the METZ but also through the transsulfuration pathway. In this pathway, vitamin B6 is required as a cofactor for the enzymatic activity of CBS. This enzyme catalyzes the conversion of homocysteine to cystathionine. In the further steps of the cycle, cystathionine is converted by cystathionine γ- lyase to cysteine and ultimately to glutathione or to sulphate. Homocysteine cannot be ingested with food but must be synthesized by metabolic processes in the body.

### 2.5. Polyamine Metabolism

Polyamines, which include putrescine, spermine, and spermidine, are synthesized in every cell type and are essential for cellular growth [20]. So far, three different sources of intake of polyamines in mammals are known: (1) They can be ingested with nutrition;(2) enzymatic synthesis under SAM consumption, hereby the intracellular polyamine concentrations are diet dependent and can be selectively reduced by inhibition of the enzymes required for their biosynthesis [21,22]; (3) production of polyamines by microorganisms [20]. Polyamines can inter alia interact with negatively charges molecules, such as RNA, DNA, phospholipids, and proteins, and thereby regulate a plethora of cellular functions [20]. Polyamine metabolism starts with the basic, non-protein genic α-amino acid L-ornithine. In the first step of the metabolic pathway, L-ornithine is converted to putrescine by ornithine decarboxylase (ODC1). This reaction represents the metabolically limiting step of polyamine metabolism, with ODC1 being the key enzyme [23]. On the cellular level, L-ornithine is synthesized from L-arginine by the enzyme arginase (ARG1). The next steps of polyamine metabolism reveal a connection between this pathway and one-carbon metabolism. Both the transfer of the aminopropyl group on putrescine to create spermidine (this reaction is carried out by the spermidine synthase (SRM)) as well as the transfer of the same chemical group on spermidine to create spermine (this reaction is carried out by the spermine synthase (SMS) require decarboxylated SAM (dcSAM) as the aminopropyl group donor. dcSAM is formed by the decarboxylation of SAM by the adenosylmethionine decarboxylase (AMD1). The intracellular levels of spermidine and spermine are mostly controlled by the export of these metabolites from the cell [20]. As an alternative, both spermidine and spermine can be processed by the enzymes spermine oxidase (SMOX) and polyamine oxidase to be converted back into putrescine [20]. An overview of polyamine metabolism is given in Figure 4.

Polyamine metabolism starts with the basic, non-protein genic α-amino acid L-ornithine. L-ornithine is formed from ARG. In the next step of the metabolic pathway, the diamine putrescine is synthesized by ODC. The next steps reveal the connection to “one-carbon metabolism” because both the transfer of the aminopropyl group by SRMto putrescine, giving rise to spermidine, and by SMS to spermidine, giving rise to spermine, require dcSAM as the aminopropyl group donor. This reaction leads to the formation of 5-methylthioadenosine (MTA). dcSAMis formed by AMD, from SAM synthesized in METZ. Alternatively, to export, spermine and spermidine can also be converted back to putrescine by the enzymes SMOX, SSAT, and PAOX.

Polyamine concentrations are regulated by two major mechanisms. For either the activity of spermidine/spermine N1-acetyltransferase 1 (SSAT), acetylation of spermine or spermidine is required. In this acetylated form, spermine and spermidine are either oxidatively cleaved into lower polyamines, or alternatively, they can be excreted from the cells into animal body fluids, such as urine or blood [20].

## 3. Influence of the One-Carbon Metabolism and the Polyamine Metabolism on Epigenetics and Cancer

### 3.1. Metabolic Influence on Epigenetics

The influence of one-carbon and polyamine metabolism on epigenetics is complex and diverse. The nutrients in our daily food intake can regulate gene expression by influencing different epigenetic mechanisms. On the one hand, there are ingredients that directly regulate the availability of SAM or are needed for the de novo synthesis of adenosine, guanosine, and thymine [12]. These nutrients are components of one-carbon metabolism, such as folate, riboflavin, betaine, serine, and methionine. On the other hand, there are ingredients that regulate histone modifications or the transcription of non-coding RNAs, such as vitamin D [24]. Other important metabolites that are not ingested with our daily food intake can also have an impact on epigenetic regulatory mechanisms. These metabolites must be synthesized by the body and include acetyl coenzyme-A (acetyl-CoA), flavine adenine dinucleotide (FAD), and α-ketoglutarate (α-KG). These metabolites can have both enhancing and inhibitory effects on the enzyme activity of histone-modifying enzymes themselves. For instance, the presence of acetyl-CoA is associated with the attachment of acetyl groups to the respective histone proteins, while increased concentrations of its deacetylated version are associated with the inhibition of histone acetyltransferases [25,26,27]. SAM has been recognized as the major player in regulating the cell’s ability to attach methyl groups to DNA as well as histones. Through the accessibility of SAM, both DNA methyltransferases (DNMTs), as well as HMTs, can be regulated [10]. The degradation product of SAM, which is synthesized during the methylation reaction, namely SAH, is responsible for the inhibition of several methyltransferases. Maruti et al. have shown that increased cellular concentrations of SAH lead to global hypomethylation [28]. Further metabolites, whose concentration changes have been linked to global hypomethylation, are HCY [29] and dcSAM [30]. Furthermore, it could be shown that the knockout of the MTHFR gene in mice resulted in significantly lower intracellular SAM and higher SAH concentrations with concomitant global DNA hypomethylation [31]. This finding was also confirmed by two studies, revealing concentration changes in the crucial SAM:SAH concentration in the blood serum by different diets [32,33]. Analogous to DNA methylation, a similar correlation has also been reported for the establishment of trimethylation at histone H4K3. This modification is also directly dependent on the availability of methionine and the intracellular production of SAM [34]. Further insight into the importance of our nutrition, physical activity, and stress was raised by a recently published study [35]. In this study, the participants received a dietary intervention with nutrients that are known to regulate either one-carbon metabolism, DNMTs, modulators of demethylation, or histone modification. Additionally, the participants had to perform physical exercise for 30 minutes per day at least 5 days a week, as well as breathing exercises to reduce stress levels. These lifestyle changes reduced the biological age, as measured by the Horvath DNAm Age clock [36], by 3 years on average compared to the control group. This direct interaction between the availability of metabolites and the activity of the epigenetic modulators has also been proposed in the “nutrient sensing model” [37]. A good example of this is the dependency of the methylation reactions of folate and methionine. Both can be ingested with food and 50% of it will be directly converted into SAM [18]. In this context, our research group was able to link the genome-wide demethylation of long interspersed nuclear elements 1 (LINE1) retrotransposons in cell-free DNA of human blood to the aging process, presenting a new biomarker of aging [38]. Taken together, these findings highlight the importance of proper supplementation of one-carbon metabolites by our daily food intake for the ability of the cell to apply methyl groups and thereby maintain cellular function and prevent or slow down the aging process. 

In the case of DNA and histone demethylation, especially α-KG plays a central role as a co-substrate and activator of Jumonji c-domain-containing HDMs and the members of the ten-eleven translocation family (TET). Metabolites that are structurally closely related to α-KG are succinates, fumaric acid, and 2-hydroxyglutarate, and these have been reported to play an inhibitory role for the before-mentioned enzymes [39,40,41]. Another layer of regulation of the epigenetic regulatory mechanisms is added through the local production of the metabolites in the nucleus [37]. Therefore, the enzymes which synthesize the respective metabolites are directly recruited to specific sites of the chromatin where the modification should be added. An example of such a local production is the recruitment of methionine adenosyltransferase isoform type 2 (MAT2A). Through direct protein–protein interactions with the transcription factor MafK, it is linked to a specific sequence in the genome and allows the local synthesis of SAM [42]. Another complex that synthesizes metabolites directly at required sequences of the genome is the so-called “serine-responsive SAM-containing metabolic enzyme complex” (SESAME). It has been shown that this complex interacts with the Set1 methyltransferase complex and supplies it with SAM which is required for the establishment of trimethylation at histone H3K4 [43]. Furthermore, evidence for the involvement of the local synthesis of metabolites and DNA repair mechanisms has been suggested [44]. 

### 3.2. Metabolic Influence on Cancer Development and Progression

Since cancer cells are a highly proliferative cell type, an elevated level of exogenous supply of lipids, amino acids, and carbohydrates is needed to maintain the high proliferation rates and thereby cellular survival. Therefore, cancer cells must adjust their metabolism in oxygen and nutrient supply. In the mid-1920s, the initial evidence for altered metabolic processes in cancer cells was provided by Otto Warburg [45]. With the “Warburg effect”, he described the metabolic shift in cancer cells from oxidative phosphorylation to an increased rate of aerobic glycolysis, to produce significantly higher amounts of ATP [46]. In the mid-1940s, researchers started to report the influence and alteration of one-carbon metabolism in cancer cells. It was found that a low folate diet was able to reduce the number of cancer cells in the blood of children suffering from acute leukemia [47]. This discovery led to the introduction of so-called antifolate drugs that are in clinical use to treat various cancer entities [48]. As examples of such drugs, methotrexate and fluorouracil are the most notable, which both act as inhibitors either of THF or thymidine synthesis [49,50]. THF is needed for the remethylation of MET in the METZ and for the synthesis of the purine base thymine. Since cancer cells have an increased need for nucleotides by their high proliferation rate, also the need for metabolites of one-carbon metabolism and especially THF is increased. By the administration of these medications, THF and thymidine synthesis is restricted to such an extent, that uracil instead of thymine is incorporated into the DNA, which prevents proliferation [51,52]. Interestingly, another study reported higher DNAmethylation levels for patients suffering from colorectal cancer who received a diet with more than 400 µg of folate compared to patients receiving less than 200 µg [53].

In addition to this direct influence of the metabolite processes on the availability of the “raw materials” of one-carbon metabolism, the influence of the metabolites can also be regulated by genetic or epigenetic mechanisms. For instance, in different cancer entities that harbor a mutation in either the isocitrate dehydrogenase (NADP (+1)) (*IDH1*) or NADP(+2) (*IDH2*) locus, the first oncometabolite (D)-2-hydroxygluterate (D-2HG) was identified. The mutations were associated with the catalyzation of D-2HG instead of α-KG from isocitrate. D-2HG was found to inhibit the function of DNA and histone demethylases and was associated with altered epigenetic regulation, collagen synthesis, and cell signaling [54]. Accordingly, to the inhibition of demethylases, the cells harboring the *IDH1* mutations were found to have increased DNA and histone methylation signatures, which were associated with altered expression profiles [55,56]. 

Another gene of one-carbon metabolism that is transcriptionally inactivated by mutation in human cancers is *MTHFR*. The contribution of MTHFR to tumorigenesis has been documented in the case of colon [57], pulmonary [58], and gastric carcinoma [59]. For this gene, the transition from cytosine to thymine at position 677 has been reported to decrease the transcription level by 70% in the homozygous and by 40% in the heterozygous case, and both situations were associated with global DNA hypomethylation [60]. Another common mutation of the *MTHFR* gene is the transversion of adenine to cytosine at position 1298. This mutation was also found to be associated with a decrease in enzyme activity and a 4% decrease in the LINE-1 methylation level [61]. The allele variant A1298C is also associated with a 4.76-fold increased risk of developing bladder cancer [62]. Furthermore, our own study reports a hypermethylation at the 5′-regulatory region of the *MTHFR* gene in early-stage urothelial carcinoma [63]. We concluded that the hypermethylation within the *MTHFR* gene could cause transcriptional silencing and therefore alter the crucial SAM:SAH ratio, since the methylation of the *MTHFR* gene was found to be inversely correlated with gene expression [64].

Our own study also revealed a significant hypermethylation within the *AHCY*5′-regulatory region in the early stages of urothelial carcinoma [63]. An impairment of this enzyme is associated with an increase in the cellular SAH concentration, which can ultimately result in the inhibition of methyltransferases and is associated with hypomethylation [65]. The downregulation of AHCY was also found to be associated with the promotion of oncogenesis by protecting the affected cells from p53 and p16ink4-induced cell cycle arrest [66]. This study also showed that in 50% of all examined cancer entities, the mRNA levels of *AHCY* were found to be downregulated. Furthermore, the knock-down of AHCY and depletion of adenosine werelinked to inducing DNAdamage and cell cycle arrest [67].

As already mentioned, dcSAM is a potent inhibitor of methyltransferases and is heavily required for the synthesis of spermidine and spermine. The rate-limiting reaction for this metabolic pathway is the decarboxylation of ornithine to form putrescine whichis carried out by ODC1. This enzyme is expressed in almost all tissues and its high fluctuation in intracellular concentration enables the cells to adapt to changing environmental and metabolic conditions. In 1997, Heljasvaara et al. demonstrated that even mice overexpressing ODC1 were able to maintain their polyamine homeostasis [68]. Increased polyamine synthesis has been linked to improving cancer survival and growth, with overexpression of ODC1 being associated with breast, lung, colon, prostate, pancreatic cancer, and others [69]. For other cancer entities including bladder cancer and oral cavity carcinoma, a downregulation of ODC1 by DNA hypermethylation and subsequently global demethylation has been reported [70,71]. Especially, the transcriptional downregulation of ODC1 by siRNA was shown to lead to hypomethylation of LINE-1 retrotransposons, to induce their transcriptional activity and DNA double-strand breaks in short-term cultivated primary bladder cells [70]. This activation of LINE-1 retrotransposons has been recognized as a hallmark of the early stages of urothelial carcinoma [72] and led to the development of the PrimeEpiHit hypothesis as the initial part of the etiology of urothelial carcinoma as reported [63].

## 4. Nutrition and Plant-Extracted Compounds Influencing the One-Carbon- and Polyamine Metabolism

### 4.1. Nutrition

Many key components necessary to supplement the cell with the basic building blocks to carry out all epigenetic modifications can be obtained through daily nutrition. This includes betaine, folate methionine, serine, and most polyamines. Through the digestion of food containing these components, the crucial SAM:SAH ratio within the cells can be modulated and this directly influences, for instance, the cells’ ability to apply methyl groups to the DNA double-strand or synthesize nucleotides necessary for cellular replication. Since the aging process and one of the most common aging [38] associated diseases, namely cancer, have both been linked to the occurrence of global DNAhypomethylation [73], the proper supplementation of methyl groups inour daily nutrition seems to be essential for a healthy and deaccelerated aging process. This association between nutrition and aging has also been observed in a study carried out by Quach et al., in which a reduction in DNA methylation age was observed in individuals consuming a specific diet [74]. Despite nutrients that directly influence the SAM:SAH ratio, there are several other molecules in our food that can affect (a) demethylation by being substrates or cofactors of the ten-eleven translocation demethylases (TET) such as alpha-ketoglutarate, vitamin C and A, (b) modulators of the DNA methyl transferases, such ascurcumin or epigallocatechin gallate (EGCG), rosmarinic acid, quercetin, and luteolin. A full list of nutrients containing key components of one-carbon and polyamine metabolism isgiven in Table 1. Furthermore, there is increasing evidence that the composition of our gut bacteria has an important effect on one-carbon metabolism. For instance, *Lactobacillus plantarum* has been shown to be able to produce folate if para-aminobenzoic acid (PABA) is available [75] and that this metabolic process is able to influence gene expression [76]. Therefore, probiotic drinks are under consideration for the treatment of several diseases including diabetes [76].

Further complexity into the uptake of metabolites affecting methyl group metabolism is added when considering sex-specific differences in the availability of key metabolites caused by sex hormones. In mammals, it could be shown that pre-menopause females exhibit lower concentrations of HCY [83] and SAM [84], while showing higher levels of betaine [85] in the blood plasma. Moreover, four key enzymes of one-carbon metabolism show sex-specific differences, namely BHMT, MTR, MTHFR, and SHMT [86]. While BHMT and MTHFR show downregulation in females compared to males, MTR and SHMT are upregulated with consequences for the concentration levels of betaine, HCY, and SAM. For instance, an increased activity of MTHFR and BHMT in response to testosterone has been reported [87]. Males have approximately 19 times more testosterone in their blood compared to females and it was shown by Schwahn et al. that the exchange of testosterone for estrogen in females reduced the expression level of BHMT by 40% [88]. Interestingly, several studies concluded that alcohol consumption is able to decrease the levels of testosterone in humans [89,90,91]. Furthermore, testosterone levels can be highly regulated by nutrition. For instance, individuals consuming a diet containing ~20% fat compared with a diet containing ~40% fat showed significantly lower levels of testosterone in the blood [92,93,94]. A similar observation was also reported for people eating a vegetarian diet, since this form of nutrition is also associated to contain fewer fatty acids [95,96,97,98]. Besides nutrition, high-intensity resistance exercises have also been shown to increase testosterone levels in humans [99,100,101]. However, testosterone not only affects enzymes of methionine and folate metabolism but also polyamine metabolism. Jotova and colleagues showed that testosterone can upregulate the enzyme activity of ODC1 up to 4 h and 12 h after hormonal treatment. Thereby, the treatment led to a 1.6-fold increase in ODC1 as well as an increase inthe intracellular concentration of spermidine and spermine after 4 h and putrescine and spermine after 12 h, by 2.2- and 2.6-fold and 1.4- and 1.5-fold, respectively [102]. This crucial interplay between testosterone and polyamine metabolism was also highlighted in the murine kidney. Levillain et al. injected pharmacological and physiological doses of testosterone into female and castrated male mice and found an upregulation of arginase II and ODC, while ornithine aminotransferase was found to be downregulated [103].

### 4.2. Naturally Existing Plant-Extracted Compounds with Impact on DNA Methylation

DNA methylation alterations successively occur during aging and profound changes persist in various age-related diseases, e.g., cancer. Few bioactive food compounds have been well described to either modulate DNA methylation by impacting the S-adenosylmethionine/S-adenosylhomocysteine ratio, or by directly affecting DNA methyltransferases [104]. Thus, it is assumed that future work may identify the nutritional measures contributing to DNA methylation patterns of healthy aging. A category of compounds naturally found in plant foods such as fruits, vegetables, herbs, spices, tea, dark chocolate, and wine is known as polyphenols. Polyphenols can act as antioxidants. They are thus able to neutralize harmful free radicals that would otherwise damage cells and increase the risk of diseases such as cancer [105]. Currently, 8000 types of polyphenols have been identified. These are divided into four main groups: flavonoids, phenolic acids, polyphenolic amides, and other polyphenols [106,107]. The number of polyphenols in foods depends on their origin, ripeness, transportation, preparation, and farming. Flavonoids are about 60% of all polyphenols, such asquercetin, kaempferol, catechins, and anthocyanins. Apples, onions, dark chocolate, and red cabbage possess these types of flavonoids. Bioactive dietary components can reactivate tumor-suppressor genes by reversing aberrant DNA methylation patterns. Thus, they have a high potential to act against different types of cancer [108]. Other phenolics include for instance resveratrol, which is found in red wine, ellagic acid in berries, curcumin in turmeric, and lignans in flax seeds, sesame seeds, and whole grains. Curcumin (diferuloylmethane), for instance, a component of the golden spice Curcuma longa, commonly known as turmeric, has been reported, beyond possessing many other epigenetic effects [109], to confer DNA hypomethylation, presumably by covalently blocking a catalytic center of DNMT1 [110]. It is thought that this mode of action is involved in curcumin’s capability to act as a powerful, chemoprotective anti-cancer agent, as demonstrated by various studies [111]. This is analogous to resveratrol from grapes, mulberries, apricots, pineapples, and peanuts with a substantial anti-cancer property [24], showing weak inhibition of DNMT activity to inhibit methylation [112], the potential to restore LINE-1 methylation levels [113], and to modulate expression levels of tumor suppressors [114]. Finally, genistein, a phytoestrogen from soybeans, has been shown to have a strong dose-dependent inhibition of DNMT activity [109] and to promote promoter demethylation and reactivation of tumor-suppressor genes in diverse cancer cell types [115,116]. Many other naturally existing plant-extracted compounds with evidence oftheir impact onepigenetic mechanisms and in particular DNA methylation are described. However, future research in this field will determine which application protocols of single or combinations of these naturally existing compounds will optimize the aging process and minimize the likelihood of cancer.

## 5. Conclusions

More and more it is evident that epigenetic mechanisms, especially DNA methylation, function as distinct genome usage orchestrators which have been shaped by direct interaction with environmental conditions to contribute to the adapted cell types. However, methyl group metabolic pathways acting in every single cell of our organism are required to translate one exogenous noxa, namely nutrition, into an appropriate methylome. Aroundabout 20 million methyl groups are necessary during every cell division. Thus, it should be inferred that deterioration and inefficiency of these key methyl group metabolic pathways may contribute to DNA methylation aberrations as we observed in dedifferentiation processes, aging, and cancer initiation and justifiably assume their causal role. Therefore, more research will be needed in this area to understand how we may be able to counteract this.

## Figures and Tables

**Figure 1 ijms-23-08378-f001:**
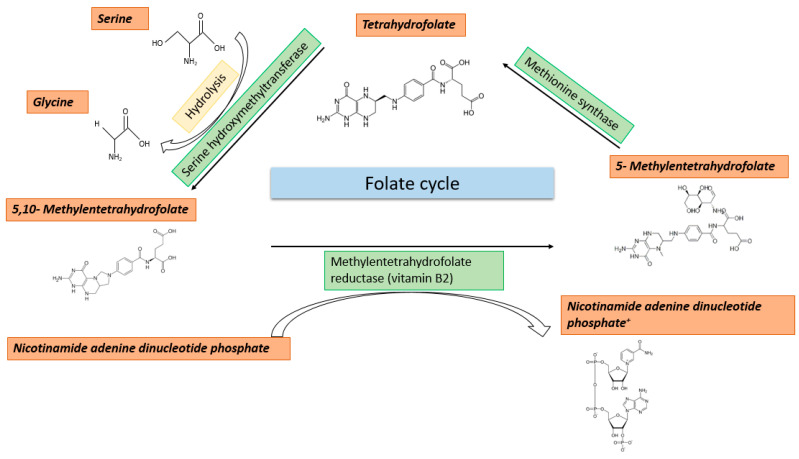
The folate cycle.

**Figure 2 ijms-23-08378-f002:**
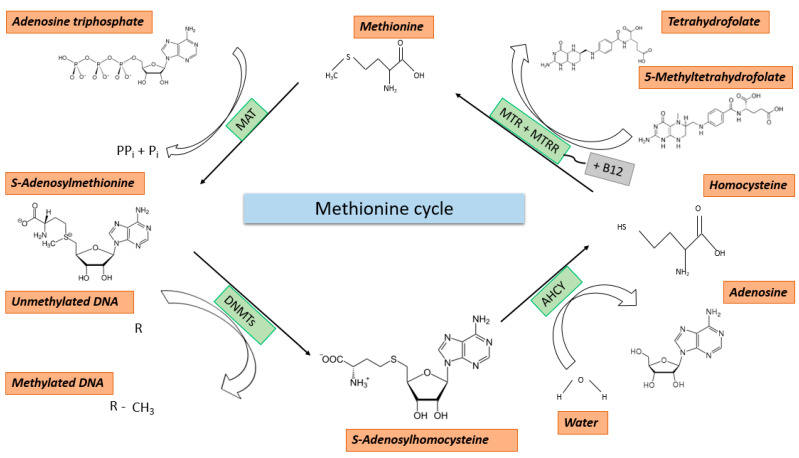
The methionine cycle.

**Figure 3 ijms-23-08378-f003:**
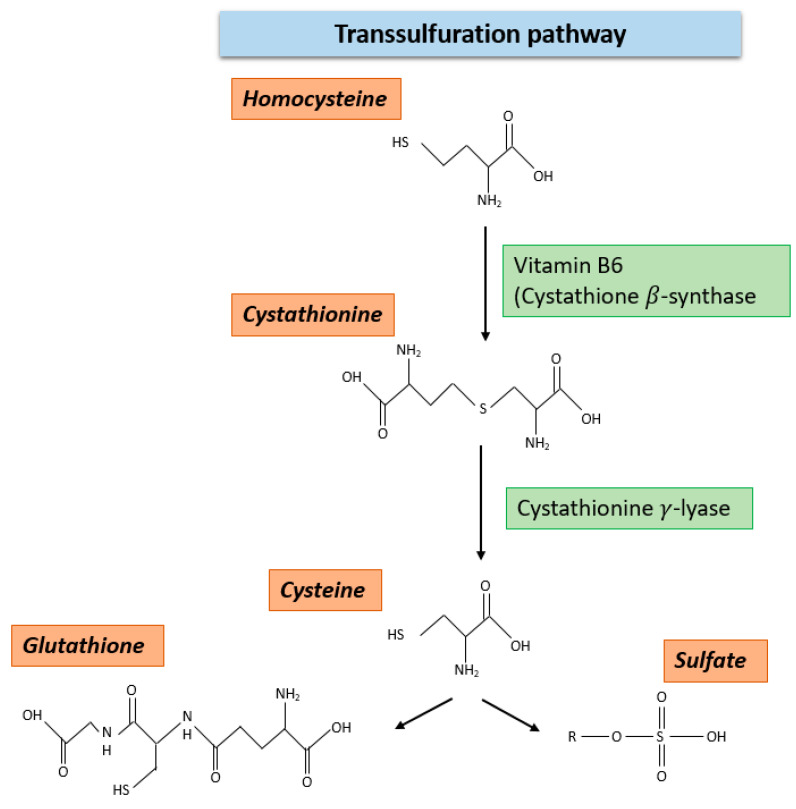
The transsulfuration pathway.

**Figure 4 ijms-23-08378-f004:**
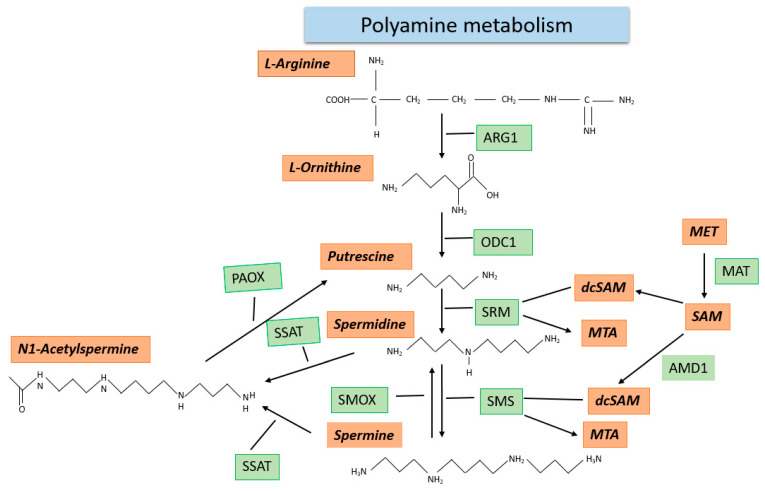
Polyamine metabolism.

**Table 1 ijms-23-08378-t001:** Food containing nutrients affecting epigenetics.

Molecule	Food	µg/100 g	Recommended Dietary Intake
Folate	Chicken liver	578	330 μg/day [77]
	Calf liver	331	
	Peanuts	246	
	Sunflower seed kernels	238	
	Lentils	181	
	Chickpeas	172	
	Asparagus	149	
	Spinach	146	
	Lettuce	136	
	Peanuts (oil roasted)	125	
	Soybeans	111	
	Broccoli	108	
	Walnuts	98	
	Peanut butter	92	
	Hazelnuts	88	
	Avocados	81	
	Beets	80	
	Kale	65	
	Bread	65	
	Cheese	20–60	
	Cabbage	46	
	Red bell peppers	46	
	Cauliflower	44	
	Chicken eggs	44	
	Salmon	35	
	Tofu	29	
	Potatoes	28	
	Chicken	12	
	Beef	12	
	Yoghurt	8–11	
	Pork	8	
	Milk	5	
	Butter	3	
		**mg/100 g**	
Arginine	Pumpkin seeds	5353	20 grams per day [78]
	Peanuts (roasted)	2832	
	Pine nuts	2413	
	Walnuts	2278	
	Peas (dried)	2278	
	Chicken breast (raw)	1436	
	Pork (raw)	1394	
	Salmon (raw)	1221	
	Buckwheat grains	982	
	Egg	820	
	Wheat flour	642	
	Rice	602	
	Corn flour	345	
	Milk	119	
Betaine	Quinoa	630	6 mg/kg body weight per day in addition to the intake from the background diet [79]
	Wheat germ	410	
	Lamb’s quarters	330	
	Wheat bran	320	
	Canned beetroot	260	
	Dark rye flour	150	
	Spinach	110–130	
	Red wine	0.76	
	Fish (shrimp)	0.75	
	Fish (tuna)	0.75	
	Fish (salmon)	0.35	
	White wine	0.12	
	Grapes	0.11	
Epigallocatechin gallate	Green tea (brewed)	70	107 to 856 mg/day [80]
	White tea (brewed)	42.45	
	Black tea (brewed)	9.36	
	Green tea	3.96	
	Pecans	2.3	
	Hazelnut	1.06	
	Cranberries	0.97	
	Blackberries	0.68	
	Raspberries	0.54	
	Black tea	0.51	
	Pistachios	0.4	
	Plums	0.4	
	Peaches	0.3	
	Apples	0.24	
Glutamic acid	Wheat flour	4328	
	Peas (dried)	4196	
	Chicken breat (raw)	3458	
	Beef (raw)	3191	
	Salmon (raw)	2830	
	Walnuts	2816	
	Egg	1676	
	Rice	1618	
	Corn flour	1300	
	Milk	687	
	Tomato puree	685	
Luteolin	Juniper berries	69.05	
	Paprika (green)	4.71	
	Celery hearts (green)	3.5	
	Artichokes	2.3	
	Chicorée	2.08	
	Lemon	1.9	
	Pumpkin	1.63	
	Grapes (red)	1.3	
	Kohlrabi (raw)	1.3	
	Parsley (fresh)	1.09	
	Paprika (yellow)	1.02	
	Kiwi	0.74	
	Paprika (red)	0.61	
Quercetin	Capers (raw)	234	Daily consumption of 25–50 mg [81]
	Capers (canned)	173	
	Lovage leaves (raw)	170	
	Buckwheat seeds	90	
	Dock-like sorrel	86	
	Radish leaves	70	
	Carob fiber	58	
	Dill	55	
	Cilantro	53	
	Hungarian wax pepper	51	
	Fennel leaves	49	
	Onion (red)	32	
	Radicchio	32	
	Watercress	30	
	Kale	23	
	Chokeberry	19	
	Bog blueberry	18	
	Cranberry	15	
	Lingonberry	13	
	Plums (black)	12	
Serine	Peanuts	1862	
	Cheese (emmentaler)	1749	
	Soybeans	1690	
	Cheese (gouda)	1570	
	Lima beans	1520	
	Lentils	1510	
	Fish (plaice)	1210	
	Fish (tuna)	1050	
	Bacon	1020	
	Walnuts	898	
		**g/100g**	
Methionine	Egg (white, dried, powder)	3.204	19 mg/kg body weight/day [82]
	Sesame seed flour	1.656	
	Brazil nuts	1.124	
	Cheese (parmesan)	1.114	
	Hemp seed	0.933	
	Soy protein concentrate	0.814	
	Chicken	0.801	
	Fish (tuna)	0.755	
	Beef	0.749	
	Bacon	0.593	
	Chia seed	0.588	
	Beef	0.565	
	Pork	0.564	
	Soybeans	0.547	
	Wheat germ	0.456	
	Egg (cooked)	0.392	
	Oat	0.312	
	Peanuts	0.309	
	Chickpea	0.253	
	Corn (yellow)	0.197	
	Almonds	0.151	
	Beans (pinto, cooked)	0.117	
	Lentils (cooked)	0.077	
	Rice (brown, cooked)	0.052	
		**mg/kg**	
Spermidine	Wheat germ	243	
	Soybean (dried)	207	
	Cheese (cheddar)	199	
	Mushroom	89	
	Rice bran	50	
	Chicken liver	48	
	Green peas	46	
	Mango	30	
	Chickpea	29	
	Cauliflower (cooked)	25	
	Broccoli (cooked)	25	

## Data Availability

Not applicable.

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
