# Peer review of "Methyl Group Metabolism in Differentiation, Aging, and Cancer"

_ijms, 2022, doi:10.3390/ijms23158378_

Round 1

Reviewer 1 Report

Erichsen et al. provide a concise overview of the diverse metabolic pathways involved in the maintenance and recycling of methylgroups within  mammalian cells. They discuss these pathways in physiological conditions and present how perturbations of these cycles could contribute to aging  and/or aging-related disorders. 

The topic is of extreme interest, but the review is difficult to read. The flow of the text is not always clear and repetitions are abundant. The writing appears disorganized and confusing. Extensive proofreading is therefore needed before the manuscript could be considered for publication.

Specifically, the authors should address the following points:

1) A figure should be added outlining the two major metabolic pathways regulating the supplementation of methyl  groups within a mammalian cell (line 77)

2) The one carbon metabolism should be broken in subparagraphs : folate cycle, methionine cycle etc;

3) Figure 1: The schematic is incomplete and there are missing information. Also 5.10-Methylen-THF is converted into 5-methyl-THF by the methylenetetrahydrofolate reductase (MTHFR), please correct;

4) All the figures should be improved and the font size increased;

5) Abbreviations should be spelt out in the main text, and not in the legends.

6) It could be easier to discuss the two metabolic pathways (one carbon and polyammine) in healthy, aging and cancer cells, respectively, without merging them in one paragraph.

7) Please move the Table before the conclusions.

Reviewer 2 Report

To a different extent, and varying in terms of incidence among the different living organisms, nearly all biological macromolecules (e.g., nucleic acid, lipids, sugars, and proteins) are methylated. The review-manuscript titled "Methylgroupmetabolism in differentiation, aging and cancer" aims to provide a compact overview of the diverse methyl group metabolic pathways operating in each cell. Additionally, the authors highlighted the association between the accelerated aging process, the development of aging-related diseases, and the deterioration of the methyl group metabolic pathways. Eventually, they discuss how the diet might interfere with the metabolism of methyl groups in human beings.  The paper will be of interest and once the following minor remarks will be fixed it will be suitable for publication.   Minor issues
  1. In their manuscript the authors mostly focus their attention on the nucleic acid (i.e., DNA) methylation, without considering significantly any other macro-molecule. This oversight should be amended throughout the whole manuscript.
  2. Figure 2 should be edited , because there is a formula in which it appears that a C atom has more bonds than expected. More in details I refer to that part of the figure labeled as "Methylated DNA". Please increase slightly the font used to write the atoms because as a such they are barely readable.
  3. The authors should at least briefly discuss the players and the molecular mechanisms underlying the polyamine cellular export (i.e. spermine lines 183-184).
  4. Dozens of typos are scattered throughout the manuscript (e.g. line 170 "Polyamine can inter alia interact..."; table 1, section betaine "red whine" and "white whine"; etc...) and they should be all carefully edited.
  5. I would suggest reorganizing the table by trying to cluster the different molecules according to their class/chemical properties (e.g., all the aminoacids should be listed below, starting with methionine, and not as sparsely as they are now). Eventually, at the end of the table, there should be folate because it is the only one in which the amount is expressed in micrograms when compared to all the others. 
  6. Throughout the whole manuscript, including the table, the numbers should be carefully edited ("." for decimal;  "," non-decimal). I am not fully convinced that currently only 8 types of polyphenols are known (line 411).
  7. The references should be carefully revised (i.e., ref #2) and formatted according to the journal guidelines.

Reviewer 3 Report

1.     Can “methylgroupmetabolism” be one word? I don’t think to and am wondering why the authors decided to make it one word. And it’s in the title which seems an odd place to make a mistake.

2.     Abstract is very poorly written, please rewrite.

3.     Should it be 5.10 or 5,10 in section 2.2 on page 2?

4.     in the folate and methionine cycle also the contributions for the purine synthesis [12]. Correct this.

5.     Vitamin names should be upper case on page 9 line 362.

Round 2

Reviewer 1 Report

The manuscript is now suitable for publication

Reviewer 3 Report

Thanks for addressing my comments.